# Screening of Patient Impairments in an Outpatient Clinic for Suspected Rare Diseases—A Cross-Sectional Study

**DOI:** 10.3390/ijerph19148874

**Published:** 2022-07-21

**Authors:** Christoph Gutenbrunner, Joerg Schiller, Vega Goedecke, Christina Lemhoefer, Andrea Boekel

**Affiliations:** 1Department of Rehabilitation Medicine, Hannover Medical School, 30625 Hannover, Germany; gutenbrunner.christoph@mh-hannover.de (C.G.); schiller.joerg@mh-hannover.de (J.S.); 2Clinic for Nephrology, Hannover Medical School, 30625 Hannover, Germany; goedecke.vega@mh-hannover.de; 3Centre for Rare Diseases, Hannover Medical School, 30625 Hannover, Germany; 4Jena University Hospital, 07743 Jena, Germany; christina.lemhoefer@med.uni-jena.de

**Keywords:** rare disease, rehabilitation, patient needs, occupation, pain

## Abstract

Background: Most rare diseases are chronic conditions with variable impairment of functionality, which can result in a need for rehabilitation. To our knowledge, there are no systematic studies on the rehabilitation needs of patients in centres for rare diseases in the literature. Our hypothesis is that participation of these patients is so limited that there is an increased need for rehabilitation. For this reason, a survey on the need for rehabilitation was carried out in all patients presenting to the centre for rare diseases, in order to assess the need for rehabilitative measures to counteract disturbances in activity and participation. Methods: A cross-sectional study was performed to collect data using a written questionnaire from December 2020 to June 2021, including patients presenting personally in the center for rare diseases. Results: Nearly 70% of the participants assessed their own ability to work as critical. Of those surveyed, *n* = 30 (44.9%) had PDI total ≥ 33 points and, thus, a clear pain-related impairment. Conclusion: The results show functional restrictions in the areas of mental well-being and activity. As expected, the health-related quality of life is reduced as compared to healthy people. Almost half of the participants reported significant pain-related impairments, however, only 9% of all respondents stated that they had received appropriate pain therapy. The results show the need for rehabilitation-specific skills in the care and counseling of patients with rare diseases.

## 1. Introduction

Rare diseases are defined by the prevalence of a defined health condition within a population. However, such thresholds vary in different countries. In the US, the term rare disease is applied to a disease affecting fewer than 64 cases per 100,000 people, while in Sweden it is applied to a disease affecting fewer than 21 cases in 100,000 people [1]. The definition of the European Commission of Public Health includes whether a disease is “life-threatening or chronically debilitating” and “of such low prevalence that special combined efforts are needed to address them” [2]. This leads to the concept of orphan diseases, meaning that a low disease prevalence prevents efforts to develop appropriate interventions, since the market is too small (neglected diseases) [1]. Many rare diseases have a genetic background, but there are also many other rare diseases such as autoimmune and infectious diseases with chronic sequelae. Lists of orphan diseases include around 5000–8000 diagnoses [3].

Patients with rare diseases often experience long delays in reaching a diagnosis of their respective disease and, therefore, affected patients may not get any appropriate treatment until a definite diagnosis is made [4]. The duration until a rare disease is diagnosed ranges from 2.3 to 11.1 years [5]. In Germany, like in many other countries, centres for rare diseases were established in the past decade [6]. Type A centres are reference centres for rare diseases and offer non-disease specific structures, e.g., for the treatment of patients with unclear diagnoses. Patients can be referred by their physicians to a reference centre if a rare disease is suspected [6]. However, rehabilitation experts are generally not included and, hence, an assessment of rehabilitation needs is not provided. However, from clinical experience, patients in specialised diagnostic services for rare diseases often have a long history of disease and often experience a strong impact on quality of life, activities, and participation. The most common diseases in this centre were muscle and joint problems, neurological symptoms, fatigue syndrome, and gastrointestinal problems as well as pathological laboratory findings of blood count, liver values, and vitamin or iron deficiencies. According to the best of our knowledge, there are no studies on the level of functioning and/or rehabilitation needs in patients with suspected rare diseases.

As a first approach, to get an insight into the frequency, nature, and intensity of limitation in functioning and the resulting rehabilitation needs, a cross-sectional pilot study was performed in the Centre for Rare Diseases (ZSE) at Hannover Medical School (MHH). Patients were referred by their physician to the ZSE. After the centre‘s experts checked the referral, including a specific patient questionnaire and previous medical investigations, the patient was discussed in an interdisciplinary board with experts from selected medical specialties. Here, a decision was made whether the patient receives an appointment at MHH or advice regarding which specialist investigations should be performed otherwise.

It is our hypothesis that in this patient population with suspected rare diseases, activity and participation are so limited that there is an increased need for rehabilitation. However, it is not yet clear in which areas of life these needs are particularly pronounced. Therefore, all patients personally presenting to the ZSE were asked to fill in a standardised questionnaire about their rehabilitation needs. This survey aims to assess the need for rehabilitative measures.

## 2. Materials and Methods

Data collection was conducted as a cross-sectional study at one point in time. The recruitment period was from December 2020 to June 2021. In total, 100 adult patients were included, who had received an appointment at the ZSE after a file review and an interdisciplinary case conference, who also had a sufficient understanding of German. In order to avoid selection bias, a full survey was conducted, and initially the number of 100 questionnaires was considered sufficient.

Access to the ZSE is subject to a staggered standardised procedure. In order to make an appointment, a patient must be referred by the treating general practitioner or specialist. Subsequently, a detailed examination of the patient’s documents takes place in order to decide on the admission after this objective preliminary screening in an interdisciplinary case conference. This is followed by personal contact with the patient in order to arrange a definitive appointment at MHH. However, this extensive time- and personnel-intensive procedure cannot preclude that patients without a rare disease will present to the outpatient clinic. All patients who went through this procedure and were invited to a presentation appointment initially received a self-report questionnaire together with an information letter from the staff of the ZSE as part of the standardised admission process. Patients could complete the assessment immediately or at a later point in time at home and return it, if necessary. Subsequently, data entry took place at the Department of Rehabilitation.

The questionnaire inventory contained a total of 46 questions, and the respondents needed about 20 min to complete the assessment.
Sociodemographic questionsRehabilitation Goal Screening (ReGoS)Pain Disability Index (PDI)Short Form-12 (SF-12)For patients of working age: Work Ability Index (WAI)Questions about previous therapies and rehabilitation

### 2.1. Sociodemographic Indicators

Sociodemographic data on age, sex, and current employment were collected to describe the survey population.

### 2.2. Limitations in Everyday Life Due to Impairment

Using the Rehabilitation Goal Screening (ReGoS), participants were asked about the limitations they experienced in everyday life. These were plotted on a scale from 0 = no problem/no impairment to 10 = extreme problem/very severe impairment, due to the condition. The screening instrument is designed for clinical decision-making and results in a profile that depicted individual rehabilitation needs. A limitation > 5 is rated as a significant limitation of the activity by the disease.

### 2.3. Impairment Due to Pain

The Pain Disability Index (PDI) is composed of seven questions addressing different domains of daily life (family and home responsibilities, recreation, social activities, occupation, sex life, self-care, vital activities). These areas were each rated by patients on a scale from 0 = no impairment to 10 = total impairment, due to pain. This resulted in a total score between 0 and 70, from which a percentage score (a score of 70 corresponds to 100% impairment) is calculated to express the respondent’s pain-related impairment [7].

### 2.4. Health-Related Quality of Life

The Short-Form-12 (SF-12) Health Questionnaire is a disease-unspecific measurement tool to assess health-related quality of life, with patients providing self-reported information about how they feel and function [8]. The SF-12 is combined into one physical and one psychological summated scale. The scores for each scale range from 0 to 100, with the assumption made that each question on the form has equal weight [9].

### 2.5. Work Ability Index

The subjective work ability of the participants was assessed with the German version of the Work Ability Index (WAI), with 11 questions addressing seven dimensions [10,11,12]. Using cut-off values, respondents’ reported work ability is categorised as follows:
–“critical” (7–27 points)—restoring work ability;–“moderate” (28–36 points)—improve work ability;–“good” (37–43 points)—support work ability; and–“very good” (44–49 points)—maintain work ability.

### 2.6. Treatments and Rehabilitation

We collected data about the therapies received so far. Participants could select multiple response options. We asked about ambulant therapies and measures, which are prescribed by doctors, such as physiotherapy, manual therapy, or medical training therapy, as well as treatments by osteopaths and naturopaths that do not require a prescription. Other interventions such as local anesthetic injections or pain management by a pain specialist are also asked for.

### 2.7. Statistical Methods

Frequencies, mean values, standard deviations (SD), minimum and maximum values, and percentages were calculated for continuous variables for descriptive analysis of the data. Intergroup differences were calculated using ANOVA.

The decision to plan a sample size of *n* = 100 was made pragmatically, as this was an initial screening of the target group in form of a pilot study. If available, the evaluation of individual questionnaire parts was carried out according to the respective standardised evaluation instructions [7,9,10,11]. For the evaluation, SPSS 26 software was used.

## 3. Results

Of the 100 questionnaires issued, 64 were returned by post. The questionnaires contained less than 5% missing values, so no data imputation was carried out. Reasons for non-participation could not be studied due to the anonymous design and the lack of reminder management (Figure 1).

### 3.1. Sociodemographic Data

Two-thirds of the respondents were women, and the average age was 44.8 ± 14.6 years. A large majority (94%) of the respondents were of working age, of which two-thirds (68%) were currently employed (Table 1). While the average age of the male participants was 53 years, females were significantly younger with a mean age of 40 years (*p* = 0.001, Chi^2^ = 2340.2, F = 13.08).

### 3.2. Restrictions Due to the Disease–Rehabilitation Goal Screening (ReGoS)

Figure 2 shows the mean values (with error bars) of limitations due to the disease on a scale of 0–10, showing significantly higher mean values in the dimensions of activity and participation as well as psychological wellbeing. Participants were particularly restricted in the areas of occupation, energy and drive, pain, and recreational activities.

### 3.3. Workability (Work Ability Index)

Data from *n* = 60 people were available for the WAI. The majority of participants showed a critical work ability (68%), whereas 25% had a moderate and 7% reported a good work ability (Table 1).

### 3.4. Pain Disability Index (PDI)

Values were available for all 64 participants. The mean value of the PDI total was 33.2 points (SD = 16.35; min = 5; max = 65) and, thus, reached a critical value of ≥33 points on average. Of those surveyed, *n* = 30 (44.88%) had PDI totals ≥ 33 points and, thus, a significant pain-related impairment of [13,14] (Table 1).

### 3.5. Health-Related Quality of Life (SF-12)

Total scales of the SF-12 were available from *n* = 55 respondents. The physical total scale was below the mean value of 40 and was, therefore, rated as “below average” (mean = 32.3; SD = 8.3; min = 16.4; max = 54.2). With a mean value of 43.7 points (SD = 12.2; min = 22.7; max = 66.7), the mental total scale was within the SD of the norm sample and was, therefore, rated neither above nor below average (Table 1).

### 3.6. Previous Treatments and Rehabilitation

Figure 3 and Figure 4 show which and how many measures the respondents (*n* = 63) had already received by the time of the survey. On average, each respondent had undergone 5.5 rehabilitation measures (SD = 1.9; min = 3; max = 11) (Table 1) (Figure 3 and Figure 4).

Of those surveyed, three (4.8%) took part in inpatient or outpatient rehabilitation in the six months prior to the survey. Four of the respondents (6.3%) applied for rehabilitation, which was not approved. Thus, 53 (84.1%) of those surveyed stated that they had not previously applied for rehabilitation (Figure 5).

## 4. Discussion

The results of this survey of patients in the centre for rare diseases of a German university hospital show that there are functional restrictions in the areas of mental well-being and activity, which differ from the dimensions of mobility and self-sufficiency. The areas of energy and drive, professional activity, and pursuing a leisure activity are particularly restricted here. These results can be explained with the results of previous studies on depression in patients with rare diseases, due to the high psychological stress caused by many rare diseases [15,16]. As expected, the health-related quality of life is also reduced as compared to healthy people, in line with other studies [17,18]. Interestingly, among those surveyed the area of mobility was moderately restricted and the area of self-sufficiency was hardly restricted. There is no question that, especially in the field of rare diseases, there are patients with severe limitations in self-care [19] and mobility. In individual cases, a caring and supportive environment might have been established for a long time and close to home, while medical care is primarily provided by local health professionals, possibly also in the context of home visits. With these patients, the barriers to presentation in an outpatient clinic of a university clinic (travel, no home visits, long waiting times, etc.) are probably markedly greater than for patients who are less severely affected.

Almost three-quarters of the participants showed critical values and one-quarter showed moderate values in terms of their ability to work, as measured with the WAI. Only 4% showed good or very good results, reflecting the need for rehabilitation. On the other hand, the number of rehabilitation measures was very low, at 10%. This was mainly because only 16% of the respondents had applied for rehabilitation. These results are paradoxical in relation to the ascertained need for rehabilitation. Reasons could be a lack of individual willingness or ability to submit the application, or a lack of knowledge and communication about the availability of this treatment option. In addition to symptom-related rehabilitation clinics, the rehabilitation offered by pension insurance also includes medical–vocational rehabilitation programs that specifically prepare people for the return to working life. They also have a standardised testing procedure to check how long a person is able to work, depending on the ailment.

In relation to a comparison group from the outpatient clinic of the Department of Rehabilitation Medicine at the same university (*n* = 1.008, median age = 53.9 (SD = 16.2), females = 66.6%, employed = 48.3%), which answered the WAI by default (critical = 36.8%; moderate = 32.6%; good = 24.6%; very good = 6%), the respondents in the current study were twice as likely to have critical work ability. In addition, the average age of the respondents in this study was 10 years below the average of the comparison group, reflecting the high priority of restoring the ability to work and participate in the labor force [20]. The results of pain-related impairment and pain intensity are comparable to patients who were treated for chronic pain [18]. Almost half of the participants reported significant pain-related impairments, however, only 9% of all respondents stated that they had received appropriate pain therapy. If one compares these numbers with the access figures of the 2% of patients with chronic pain to pain experts in Germany, despite the high and insufficiently covered need [21,22], this represents a higher access in our patient group. The extent to which access was increased was due to the diagnosis of a specific rare disease or the commitment of the patient, since the referring physician or ZSE staff remains available. In analogy to the underutilization of pain therapy, only 21% of those questioned had access to psychotherapeutic therapies, with a corresponding reduction in psychological well-being. Here, there are reports about difficult and prolonged access routes in Germany, too [23]. Despite the complex and lengthy history of the individual diseases and the limitations in psychological well-being presented, only 30% of those surveyed stated that they actively exercised. This independent, active adaptive coping strategy is particularly useful in patients with chronic illnesses and disabilities [24]. On the other hand, dysfunctional coping strategies, with numerous changes of physicians and therapists and passive therapy strategies, also pose barriers for coping, i.e., avoiding supportive measures or not applying for rehabilitation measures [25].

The results of the survey emphasise the need for rehabilitation-specific skills in the care and counseling of patients with rare diseases. Especially in the case of the clear limitations in the areas of activity, rehabilitation-specific control, and advice, deficits for occupational-rehabilitation measures could be uncovered. Furthermore, both pain therapy and psychotherapy expertise for the care of patients with rare diseases are warranted to meet the complex needs [20].

This study has several limitations. The group of respondents is correspondingly small with 100 participants, and despite the intensive pre-screening procedure by experts from various disciplines, there is the possibility of including patients without a definite rare disease. In the future, withholding the questionnaire from the treating physician in the outpatient clinic might rule out these individual cases [26]. Another weakness is the selection of the respondents, since the focus was on a specific outpatient setting, and it can be assumed that patients who are already well-integrated into local–regional care structures did not take part. In this study, we focused on adult patients and did not include children who presented at the centre. This is a lesson learned from the pilot study. We will integrate children in the upcoming study, in order to analyze their rehabilitative needs. The ICF-based questionnaire used, ReGoS, is a pragmatic and innovative questionnaire that is used in the general outpatient setting, but validation is pending and currently being evaluated by another independent project. Nevertheless, the questionnaire clearly indicated the individual dimensions and limitations and can easily be integrated into clinical practice.

## 5. Conclusions

This study cannot fully address the care needed for individual rare diseases, but it serves to assess the rehabilitation demand and, above all, give food for thought for improving interdisciplinary treatment according to patient needs.

## Figures and Tables

**Figure 1 ijerph-19-08874-f001:**
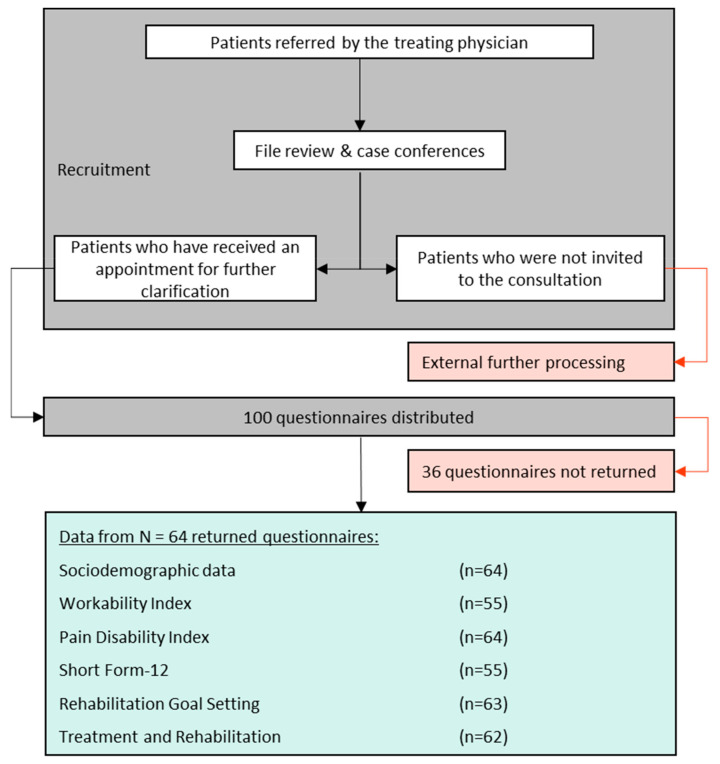
Flow chart of patient recruitment and data obtained.

**Figure 2 ijerph-19-08874-f002:**
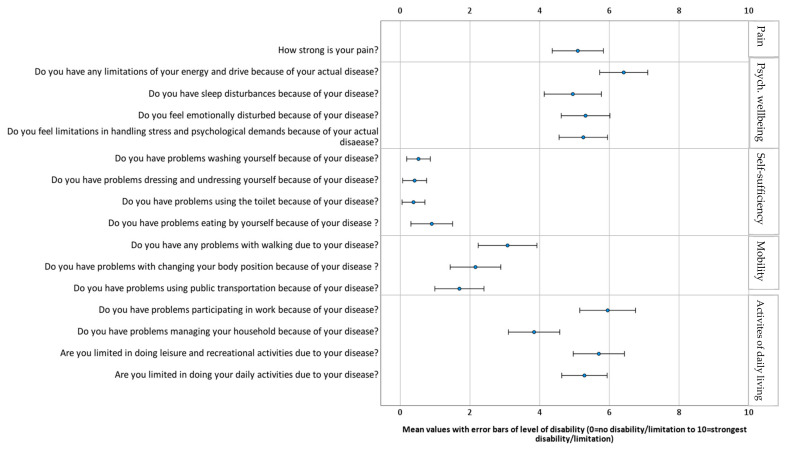
Mean values with error bars of level of disability (*n* = 63).

**Figure 3 ijerph-19-08874-f003:**
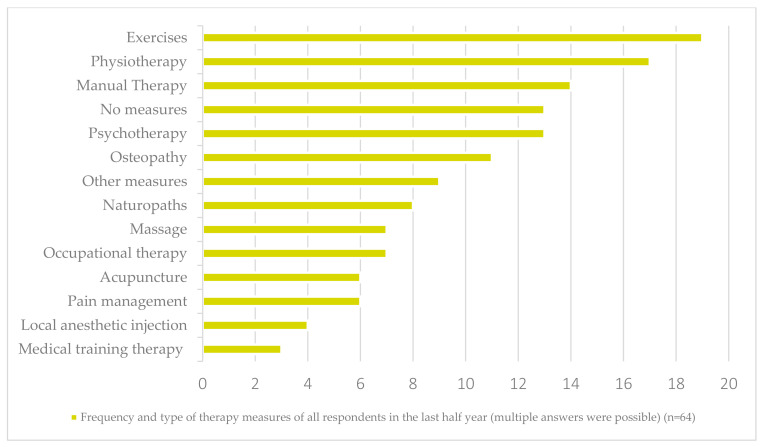
Frequencies of therapeutic interventions the respondents received during the last six months (multiple answers possible) (*n* = 64).

**Figure 4 ijerph-19-08874-f004:**
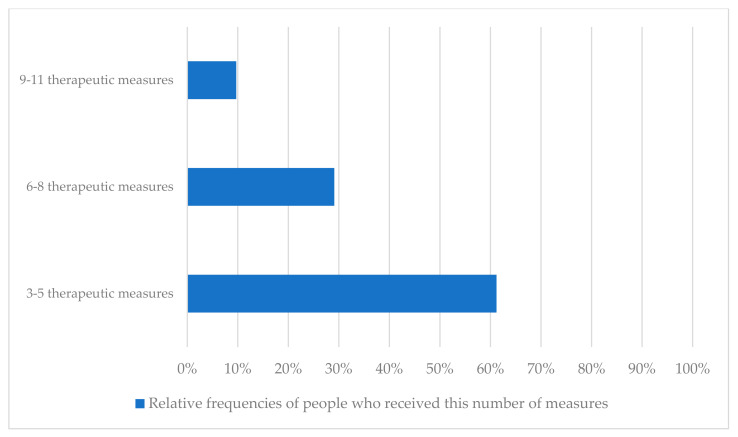
Relative frequencies of people who received 3–5, 6–8, or 9–11 therapeutic measures in the last half year.

**Figure 5 ijerph-19-08874-f005:**
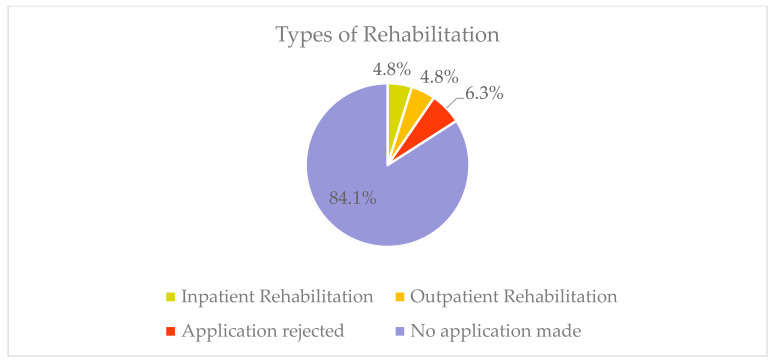
Relative frequencies of types of rehabilitation (*n* = 53).

**Table 1 ijerph-19-08874-t001:** Several parameters of the cohort including sociodemographic characteristics and data from WAI, SF-12, PDI, and rehabilitative interventions.

Parameter			Total
			Mean (SD)/%
Sociodemographics (*n* = 64)	Female		42/65.6
	Male		22/34.4
	Age overall [years]		44.8 (14.6)
	Females	Mean (SD)/%	Min/Max
	Age of females	40.4 (14.4)/	19/72
		*n*	%
	Age group 20–40	26	63.4
	Age group 41–60	11	26.8
	Age group >61	4	9.8
	Males	Mean (SD)/%	Min/Max
	Age of males	53.1 (11.2)/	33/80
		*n*	%
	Age group 20–40	3	13.6
	Age group 41–60	15	68.2
	Age group >61	4	18.2
			mean (SD)/%
Employed (*n* = 63)	Yes		40/63.5
	No		23/36.5
Work Ability Index (*n* = 60)	Critical		/68.33
	Moderate		/25.00
	Good		/6.67
	Very good		/0.00
Short Form 12 (*n* = 55)	Physical		32.3 (8.29)
	Psychological		43.7 (12.23)
Pain Disability Index (*n* = 64)			33.2 (16.35)
Number of rehabilitative interventions (*n* = 62)			5.5 (1.9)

## Data Availability

The data presented in this study are available on request from the corresponding author. The data are not publicly available due to a data protection act.

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
