# Peer review of "Screening of Patient Impairments in an Outpatient Clinic for Suspected Rare Diseases—A Cross-Sectional Study"

_ijerph, 2022, doi:10.3390/ijerph19148874_

Round 1
Reviewer 1 Report
This manuscript described a pilot study in screening the impairments in outpatient clinic for suspected rare disease, and the results suggested the need for referral/integration rehabilitation specialty into standard care of rare diseases.
Here are some points:
Introduction
The aim of the study and related background were provided.
It is of interesting that there is a reference centre for rare diseases in German to treat suspected rare disease. However, the authors should define whether this reference centre receive adult or pediatric population. Since in the latter part, only adult participants were enrolled into the study.
Method
The spectrum of suspicious of rare diseases can be broad. The affected domains are varied as well.
Were there any inclusion criteria for enrollment? If the rare disease affects cognitive or expression (language), were they eligible for survey?
It is of interest that why were these participant being recognized as suspicious rare disease. I suggested adding information of their clinical presentation (limb weakness, cognitive impairment, body structure)
Line 99
Is t Rehabilitation Goal Screening (ReGoS) a new screening tool? Is this screening being evaluated of its validity?
Line 133 the sentence needs to be revise
Figure 1 The abbreviation in the figure should be explained or avoid use the abbreviation
Table 1 The table can be simplified. The layout looks complicated.
Line 196 The figure should be Figure 3 not 2
The term used in the figure 3 is not precise, and the classification is not appropriate.
Manual therapy, exercise, massage can be part of the physiotherapy. How do you define the special treatment from each other?
Also, the term “pain therapy” is not suitable. Are you meaning medication? Interventional therapy? Exercise, physiotherapy all can be part of the treatment.
What is “infiltrations” meaning? Please consider other term
“Strength training” can be strengthening exercise or resistive exercise
Figure 4
The number of therapies that the participants received is an issue. However, the frequency and duration is another issue that clinicians being interest to know.
Results
Were the population limited to adult? From the result, all the participants were adult, which means the included of suspicious rare disease were relatively late onset or mild symptoms that the patients came for survey of the rare disease. Since the spectrum of rare diseases varies, the author should analyze and present these participants’ education status, physical impairment (by functional scale), and whether the participants were being normal performance before their referral to the centre. Though this is a cross-sectional study, more information to present the heterogeneous group characteristics is suggestion.
Line 222-224
“Only 4% showed good or very good results, reflecting the need for rehabilitation”
I suggest describe more about the rehabilitation, including intensive training for work abilities, job redesign, since most of the participants have difficulty and may not be qualified to usual work.
Line 231
What is the comparison group? Please describe more information and provide important data from the comparison group.
Line 236 “for chronic pain 18”, 18 should be removed.
Author Response
Dear Reviewer, thank you very much for pointing out some topics. It helped us to make this manuscript better.
- It is of interesting that there is a reference centre for rare diseases in German to treat suspected rare disease. However, the authors should define whether this reference centre receive adult or pediatric population. Since in the latter part, only adult participants were enrolled into the study.
- Reference centres receive referrals from general practitioners as well as specialists. Only adult patients were evaluated in the outpatient clinic for undiagnosed diseases as pediatric patients were seen in the pediatric department.
- Method
The spectrum of suspicious of rare diseases can be broad. The affected domains are varied as well. Were there any inclusion criteria for enrollment? If the rare disease affects cognitive or expression (language), were they eligible for survey?
- Inclusion criteria were presented in lines 67-72. In order to cover the broad spectrum, we have included all persons who presented at the outpatient clinic. During this time period all patients were able to communicate and answer the questionnaire. In our daily clinical practice, the receptionists or relatives, accompanying the patient, help to fill in the questionnaire. We will have to standardize this for our furthers studies. That is an important point. Thank you.
- It is of interest that why were these participant being recognized as suspicious rare disease. I suggested adding information of their clinical presentation (limb weakness, cognitive impairment, body structure)
- Their physicians recognized them as suspicious rare disease (please see lines 55-59). This was an anonymous study, so we do not have the possibility to connect to the medical record. We have also recognized this lack of information. In our next study we are going to link the medical record to the questionnaire, so we will be able to trace which symptoms and which disease were diagnosed.
- Line 99
Is t Rehabilitation Goal Screening (ReGoS) a new screening tool? Is this screening being evaluated of its validity?
- The manuscript about the REGoS tool is currently submitted for publication. As this study was supposed to be the pilot study validation projects are planned for 2023. The exact plan foresees the following steps:
- Validation and retrospective longitudinal data analysis of the ReGoS in the clinic for rehabilitation medicine.
- Retrospective longitudinal analysis of the ReGoS data from the PMR.
- Examination of the degree of goal attainment.
- Analysis of need and actual care provided
iii. Is there a relationship between rehabilitative interventions and goal attainment or changes in limitations over time?
- Validation using the Pain Disability Index.
- Investigation of the practicality and applicability of the Rehabilitation Goal Screening (ReGoS) in different settings.
- Is the ReGoS practically applicable in
- Clinic for rehabilitation medicine MHH,
- University Hospital Jena &
iii. Rehabilitation Clinic Bad Eilsen?
- To what extent are adaptations necessary?
- ReGoS results:
- Are the results similar among the settings?
- To what extent did the patient populations differ?
iii. To what extent do the ReGoS outcomes differ?
- Examine practicality in different settings through user interviews:
- Can the ReGoS be applied in all settings?
- Are the results used for common goal setting?
iii. Are there suggestions for improvement?
- Line 133 the sentence needs to be revise
- We revised the sentence.
- Figure 1 The abbreviation in the figure should be explained or avoid use the abbreviation
- We have revised the figure and removed the abbreviations.
- Table 1 The table can be simplified. The layout looks complicated.
- We choose a new layout and revised the figure.
- Line 196 The figure should be Figure 3 not 2
- We corrected the figure.
- The term used in the figure 3 is not precise, and the classification is not appropriate. Manual therapy, exercise, massage can be part of the physiotherapy. How do you define the special treatment from each other? Also, the term “pain therapy” is not suitable. Are you meaning medication? Interventional therapy? Exercise, physiotherapy all can be part of the treatment.
- That's an important note from you. In Germany, almost all of the interventions listed here have to be prescribed by a doctor, which is why these are fixed terms in this context, such as massage, manual therapy and physiotherapy. We explained it in the text (lines 136-141). The term "pain therapy" in our context is pain management by a pain specialist like a specialized physician. It can include all the therapies you mentioned, but in this case it a managed ba a specialized physician.
We have renamed the figure.
- What is “infiltrations” meaning? Please consider other term
- We used another term “local anesthetic injections”.
- “Strength training” can be strengthening exercise or resistive exercise
- That was a translation mistake. The term is “Medical training therapy” and it is a form of therapy that also has to be prescripted by the physician. We corrected the term.
- Figure 4
The number of therapies that the participants received is an issue. However, the frequency and duration is another issue that clinicians being interest to know.
- This question was asked with the intention of finding out whether the respondents have already received therapeutic care or whether they have had to manage without any therapies so far. It will now be necessary in further studies to ask how much therapy they have received. Unfortunately we have not data about the frequency.
Results
- Were the population limited to adult? From the result, all the participants were adult, which means the included of suspicious rare disease were relatively late onset or mild symptoms that the patients came for survey of the rare disease. Since the spectrum of rare diseases varies, the author should analyze and present these participants’ education status, physical impairment (by functional scale), and whether the participants were being normal performance before their referral to the centre. Though this is a cross-sectional study, more information to present the heterogeneous group characteristics is suggestion.
- Yes, the population was limited to adults (> 18 years). In this pilot study we unfortunately did not include the medical record or implemented questions about financial or educational status. Since this are important influencing factors we are going to implement them in the next studies.
- Line 222-224
“Only 4% showed good or very good results, reflecting the need for rehabilitation”
I suggest describe more about the rehabilitation, including intensive training for work abilities, job redesign, since most of the participants have difficulty and may not be qualified to usual work.
- We transcribed more about the rehabilitation (lines 252-256).
- Line 231
What is the comparison group? Please describe more information and provide important data from the comparison group.
- We added more information about the comparison group including number, age, sex, employment status and Workability Index results (lines 257-260).
- Line 236 “for chronic pain 18”, 18 should be removed.
- The “18” was a reference and was reformatted.
Reviewer 2 Report
Great paper, addressing an interesting and useful subject, rehabilitation needs.
Indeed, there are some concerns regarding number of subjects and diagnosis. I would like to have a clue about suspected diagnosis because rare diseases are a such inhomogeneous group and I find really difficult to understand the item. Maybe a description of the general population from the center could offer some ideas about diseases, and we can assume that also this sample is quite similar to it.
The methodology used is appropriate and results clear presented and statistical analysis is limited to the available data.
Your discussions and good, and comprehensive and conclusions are in line with the original hypothesis - we need a more data because rehabilitation is need and not offered.
I really look forward for the new study.
Author Response
Dear Reviewer, thank you very much for pointing out some topics. It helped us to make this manuscript better.
Maybe a description of the general population from the center could offer some ideas about diseases, and we can assume that also this sample is quite similar to it.
- Unfortunately, we have no information about the diagnoses of the patients in the study subgroup and therefore, we cannot say whether our sample corresponds to this population. We added the most common diseases in the general population of the centre (lines 52-54). We agree with you that the sample is small and this is a lesson-learned from this pilot study: we have to choose a much bigger population in the upcoming study.
Round 2
Reviewer 1 Report
I have no further comments.